# Characteristics and Challenges of Epilepsy in Children with Cerebral Palsy—A Population-Based Study

**DOI:** 10.3390/jcm12010346

**Published:** 2023-01-01

**Authors:** Ana Dos Santos Rufino, Magnus Påhlman, Ingrid Olsson, Kate Himmelmann

**Affiliations:** 1Paediatric Neurology, Queen Silvia Children’s Hospital, Sahlgrenska University Hospital, 41685 Gothenburg, Sweden; 2Epilepsy Center Frankfurt Rhine-Main, Center of Neurology and Neurosurgery, University Hospital, Goethe-University Frankfurt, 60590 Frankfurt am Main, Germany; 3LOEWE Center for Personalized Translational Epilepsy Research (CePTER), Goethe-University Frankfurt, 60590 Frankfurt am Main, Germany; 4Gillberg Neuropsychiatry Centre, Institute of Neuroscience and Physiology, Sahlgrenska Academy, University of Gothenburg, 41119 Gothenburg, Sweden; 5Department of Pediatrics, Institute of Clinical Sciences, Sahlgrenska Academy, University of Gothenburg, 41685 Gothenburg, Sweden

**Keywords:** epilepsy, cerebral palsy, children, neuroimaging, seizure outcome

## Abstract

The aim of this population-based study was to describe the prevalence and characteristics of epilepsy in children with cerebral palsy (CP), focusing on antiseizure medication (ASM) and seizure outcome. Findings were related to CP type, gross motor function and associated impairments. Data on all 140 children with CP born in 2003–2006 were taken from the CP register of Western Sweden. Medical records were reviewed at ages 9–12 and 13–16 years. In total 43% had a diagnosis of epilepsy. Epilepsy was more common in children with dyskinetic CP, who more often had a history of infantile spasms, continuous spike-and-wave during sleep and status epilepticus. Neonatal seizures, severe intellectual disability, severe motor disability and autism were associated with a higher risk of epilepsy. Many children were on polytherapy, and valproate was frequently used, even in girls. At age 13–16 years, 45% of the children with epilepsy were seizure free for at least one year. Onset after 2 years of age, female sex and white matter injury were associated with good seizure outcome. Despite the risk of relapse, reduction or discontinuation of ASM could be an option in selected cases. It is important to optimize ASM and to consider the possibility of epilepsy surgery.

## 1. Introduction

Children with cerebral palsy (CP) often have associated impairments and disorders. Epilepsy is a common comorbidity and occurs in 15–55% of children and adults with CP [1] while in the general childhood population, the prevalence of epilepsy is between 3.2 and 5.5 per 1000 in developed countries [2].

In a previous report from the CP register of western Sweden on children born in 1999–2002, 44% had epilepsy. This was a higher occurrence of epilepsy compared to previous birth-year cohorts, possibly due to an increase in cortical/subcortical lesions documented by neuroimaging [3].

Up to 50% of children with CP and epilepsy have been reported to have seizures despite antiseizure medication (ASM) [4,5]. A model has been described for predicting drug resistant epilepsy in children with CP, using four independent predictors: low Apgar score at 5 min, neonatal seizures, focal epilepsy clinically and on routine electroencephalogram (EEG) [6].

A recent study has shown that epilepsy in CP can remit with ASM in about 50% of all cases, and in up to 20% in spastic quadriplegia or hemiplegia even with previous status epilepticus or daily seizures, up to 10 years from epilepsy onset. ASM could be discontinued without relapse in 14%. In that study, older age, perinatal aetiology and improvement on EEG were favourable factors for terminating ASM [7].

The aims of the present study were to describe the prevalence of epilepsy in children with CP in a population-based cohort born in 2003–2006 from the CP register of western Sweden, to explore the epilepsy characteristics, EEG findings and treatment in relation to CP type, gross motor function and associated impairments, and explore seizure outcome at follow-up. Furthermore, we intended to compare characteristics of children with and without epilepsy and identify factors that may favour seizure remission.

## 2. Material and Methods

The medical records of all children with CP born from 2003 to 2006 in the CP register of western Sweden and currently living in the county of Västra Götaland, Sweden, were reviewed and data of the selected cohort were analyzed at two different timepoints: at ages 9–12 (data collected between July and October 2015) and 13–16 years (follow-up data collected in 2019). At the first timepoint the following data were analyzed: age of epilepsy onset, type of seizures, frequency of seizures, status epilepticus, EEG findings, use of ASM, and non-pharmacological treatment, cognitive level, and neuropsychiatric diagnosis. For the analyses at the second timepoint updated information about new cases of epilepsy, seizure freedom and type of ASM was collected. Information about cognitive function and neuropsychiatric diagnoses was obtained from another study from our centre [8].

The whole cohort comprised 140 children (61 girls and 79 boys) and was part of a larger cohort that has previously been reported [3]. Information about type and aetiology of CP, gestational age, Apgar score at 5 min, gross motor function, neuroimaging findings and history of neonatal seizures was taken from the CP register of western Sweden.

The aetiological period was defined as prenatal when the brain insult occurred during pregnancy until the onset of labour resulting in delivery, and as peri/neonatal from the onset of labour up to day 28. Data on aetiological period for this cohort have previously been published [3].

Neonatal seizures were defined as seizures occurring up to 72 h of age.

Magnetic resonance imaging (MRI) findings were classified according to the MRI classification system for children with cerebral palsy (MRICS) into five categories: maldevelopments, predominant white matter injury, predominant grey matter injury, miscellaneous and normal. The group with predominant grey matter injury was divided into: basal ganglia/thalamus lesions, cortical-subcortical lesions, and arterial infarctions [9]. Computer tomography (CT) findings were included in the absence of MRI and classified in a similar way [3,10].

The definition of CP was the one agreed upon at an international consensus meeting in Bethesda: Cerebral palsy (CP) describes a group of disorders of the development of movement and posture, causing activity limitation, that are attributed to non-progressive disturbances that occurred in the developing foetal or infant brain. The motor disorders of cerebral palsy are often accompanied by disturbance of sensation, cognition, communication, perception, and/or behaviour, and/or by a seizure disorder [11].

The CP types were classified into unilateral spastic (USCP), bilateral spastic (BSCP), dyskinetic or ataxic CP according to the Surveillance of Cerebral Palsy in Europe (SCPE 2000) [12], and according to Hagberg into hemiplegia, diplegia, tetraplegia, dyskinetic CP and ataxia [13]. To describe the gross motor function the Gross Motor Function Classification System (GMFCS) was used [14].

The 2014 definition of epilepsy according to the International League Against Epilepsy (ILAE) was used [15]. The types of epileptic seizures during the last two years were documented. Information about previous seizure types, including infantile spasms, was also registered. Seizures were classified according to the ILAE classification of seizure types in which the mode of seizure onset is subdivided into generalized, focal, and unknown [16]. The term ‘multiple seizure types’ was used when children had a combination of both focal and generalized seizures.

Seizure freedom was defined as absence of seizures (with or without ASM) for at least one year preceding data collection [17,18].

Seizure frequency was divided into daily seizures (at least once a day), weekly seizures (less than once daily and more than once a week), monthly (at least once a month but not more than once a week), seldom (less than once a month) and clusters (many seizures in clusters followed by seizure free periods). Information was taken from the medical records regarding status epilepticus which in most cases implied hospital care.

All results of EEGs were interpreted at the Department of Clinical Neurophysiology, Sahlgrenska University Hospital, Gothenburg. Results from the EEG recordings (performed before October 2015) were documented and reviewed together with an experienced clinical neurophysiologist. Epileptic discharges were divided into focal, multifocal, and generalized epileptic discharges. Information was obtained about the occurrence of hypsarrythmia and continuous spike-and-wave during sleep (CSWS), defined as an EEG pattern of almost continuous, slow (1.5–2 Hz) spike-wave during slow sleep.

Cognitive level was divided into normal (IQ > 70), mild intellectual disability (ID) (IQ 50–70) and severe ID (IQ < 50), based on neuropsychological assessment or in a few cases on clinical observation. Neuropsychiatric comorbidity referred to autism spectrum disorder (ASD) and attention-deficit/hyperactivity disorder (ADHD). In cases where diagnostic criteria were nor fulfilled, symptoms such as attention and concentration deficits, behavioural problems, routine dependency were also registered.

### Statistical Analysis

We used descriptive statistics for the comparison of groups. The chi-square test for independence was used for the association between categorical variables; if more than two categories were present within an ordinal scale, the Cochran–Armitage chi-square test for trend was used. A *p*-value of less than 0.05 was regarded as statistically significant.

## 3. Results

In the total cohort, 60/140 (43%) had a diagnosis of epilepsy. At the review of the medical records in 2015 one child with epilepsy and one without epilepsy had died, leaving 138 children (60 girls and 78 boys), 59 children with epilepsy and 79 without epilepsy. Register data on the children with and without epilepsy are presented in Table 1.

Epilepsy was more common in children born at term (χ^2^ = 6.24; *p* = 0.012), compared to children born preterm. Children with epilepsy had more often peri/neonatal aetiology (61%; 36/59) compared to those without epilepsy (44%; 35/79). Prenatal aetiology was present in 32% (19/59) of children with epilepsy and 28% (22/79) of those without epilepsy, whereas unknown aetiology was found in 7% (4/59) and 28% (22/79), respectively.

An Apgar score < 5 at five minutes was associated with later epilepsy in term-born children (χ^2^ = 6.07; *p* = 0.014). Nearly two thirds of children with a history of neonatal seizures developed epilepsy compared to one third without neonatal seizures (χ^2^ = 10.25; *p* = 0.001). Most children (72%; 31/43) with neonatal seizures were born at term, and 24 of 31 developed epilepsy. A history of neonatal seizures was more frequent in children with dyskinetic CP (63%; 15/24) than in those with other types of CP (χ^2^ = 13.30; *p* < 0.001).

The most common neuroimaging finding in children with epilepsy was predominant grey matter injury (44%; 25/59), which was mostly seen in children born at term.

BSCP was the most common type of CP in children with epilepsy, while the children with dyskinetic CP had the highest occurrence of epilepsy.

A more severe motor impairment, i.e., wheelchair ambulation, was associated with more epilepsy (73%; 29/40) than preserved walking ability (31%; 30/98) (χ^2^ = 20.36; *p* < 0.001). Most children without epilepsy were at GMFCS levels I and II.

### 3.1. Epilepsy Characteristics

Many children (37%; 22/59) had epilepsy onset during the first year of life (Figure 1).

Most children (80%; 47/59) had an epilepsy diagnosis before five years, and by the age of ten all 59 had an epilepsy diagnosis. Age at onset varied by CP type. All six children with spastic tetraplegia had epilepsy onset before one year of age, while all four children with ataxic CP developed epilepsy at a later age.

Details on epilepsy in the 59 children, by CP type, are described in Table 2. In a few cases it was not possible to determine the frequency or detailed semiology of seizures due to lack of information in the medical records.

Focal seizures were the most common seizure type in all CP types (58%; 34/59), most prevalent in USCP (79%; 11/14).

Eight children had a history of infantile spasms, all with hypsarrhythmia on the EEG. The age of onset of infantile spasms varied between 2,5 and 16 months. Six of them had dyskinetic CP. Neuroimaging showed that four of the patients with infantile spasms had cortical/subcortical injury, two had predominant white matter injury, one had lesions in the basal ganglia, and one had miscellaneous findings. One child was at GMFCS level IV, and seven were at GMFCS level V. All except one had severe ID. Seven out of eight developed other types of seizures. One child had seizures seldom, one had seizures in clusters, two had weekly and three daily seizures. One child with dyskinetic CP with a history of infantile spasms, mild ID and predominant white matter injury was seizure free without medication.

Seizure frequency varied from daily seizures (14%; 8/59) to seizures occurring less than once a month (29%; 17/59), while 25% (15/59) of the children had been seizure free for one to six years, most of whom had a history of focal seizures (Table 3).

Children with BSCP had the highest frequency of daily seizures (21%; 5/24); two of them had tetraplegia. (Table 2).

Nineteen per cent (11/59) of the children with epilepsy had a history of status epilepticus once or on several occasions: six of them had dyskinetic CP.

### 3.2. EEG Findings

Epileptic discharges were present in 88% of the latest EEGs and were mainly focal (58%) or multifocal (27%) (Table 2).

Eight children had one or several previous EEG recordings with CSWS. Five of them had dyskinetic CP and three had USCP. The neuroimaging findings were basal ganglia lesion in five children (four with dyskinetic CP and one with USCP), and cortical/subcortical injury, periventricular white matter injury and maldevelopment in one child each.

### 3.3. Epilepsy Treatment

Fifty-four of the 59 children with epilepsy had ASM in 2015 (92%). Out of 15 seizure free children, 13 were still on medication. Half of the children still on ASM had monotherapy and half had polytherapy: 17/54 (31%) had two, 9/54 (17%) three, and 1/54 (2%) four ASMs.

The most common drug used in monotherapy was valproate (14/27), followed by oxcarbazepine (6/27) and lamotrigine (5/27). There was a wide range of drug combinations as polytherapy. Valproate was also the most common drug used in combination with other drugs (18/27) such as lamotrigine, nitrazepam or levetiracetam, but many other combinations were also used. Levetiracetam was the second most used drug in polytherapy, used in different combinations in 12/27.

Valproate was a common drug in both sexes: 10/28 girls (36%) and 22/31 boys (71%) were on valproate.

Out of the 44 children who were not seizure free, nine had been referred for presurgical evaluation. Four children underwent epilepsy surgery between the age of 5 and 7 years, with different surgical procedures. At the 2-year follow-up after surgery, one was seizure free, and two had more than 75% reduction in seizure frequency. Five children had been referred for evaluation but were not accepted for surgery.

Two children had tried ketogenic diet without effect and the diet had been stopped. Vagus nerve stimulator had not been used in the studied group.

### 3.4. Follow-Up in 2019

Between 2015 and 2019 one child with USCP and epilepsy was lost to follow-up (moved abroad). There were no new cases of epilepsy, leaving 58 who were followed up.

Eleven children had become seizure free. Thus, 26 (17 girls and 9 boys) of 58 children were seizure free (45%) and had been so for 1–10 years. Nine of the 26 seizure free children were off medication. More girls than boys had become seizure free (χ^2^ = 5.52; *p* = 0.019). Children with predominant white matter injury had more often become seizure free (79%; 11/14) than children with other neuroimaging findings, distributed on all other groups (χ^2^ = 8.50; *p* = 0.004). Children with an early epilepsy onset (<2 years of age) were less likely to become seizure free than those with a later onset (χ^2^ = 8.22; *p* = 0.004). Seven of the 26 children had USCP, ten had BSCP, seven had dyskinetic CP while two had ataxic CP. Less severe motor impairment (GMFCS level I–III) was found in 17 (65%) while 9 (35%) used wheelchair ambulation. None of the 15 children at GMFCS level V and severe ID had become seizure free at follow-up.

At follow-up 48/58 (83%) were on ASM, 20 on monotherapy and 28 on polytherapy. Five out of six (83%) children with tetraplegia were on polytherapy. Valproate was still the most common drug used in monotherapy (9/20) followed by lamotrigine (5/20). The most common drug used in polytherapy was also valproate (19/28), followed by levetiracetam (13/28). Similar drug combinations were used as in 2015. Three out of 10 girls who had previously been on valproate had stopped medication and one had changed to levetiracetam, but six girls aged 13–16 years were still on valproate, in combination therapy in four and in monotherapy in two.

At follow-up at the age of 13–16 years, five more children, all with epilepsy, in total 84% (49/58) had a diagnosis of ID. The occurrence of epilepsy increased by ID (χ^2^_trend_ = 48.05, *p* < 0.001). All eight children with a history of infantile spasms had ID. Five with CSWS had severe ID, and two had a cognitive level within the normal range.

Autism was more common in children with epilepsy than in those without (χ^2^ = 10.54; *p* = 0.001), while there was no difference regarding ADHD (χ^2^ = 1,79; *p* = 0.18). Out of the seizure free children 73% (19/26) had ID, 10 with mild and 9 with severe ID, leaving seven (27%) with an intellectual level within the normal range. Eleven children had autism and 14 ADHD.

In children who were not seizure free, 94% (30/32) had ID, 4 with mild and 26 with severe ID. Eleven children had autism and six had ADHD.

## 4. Discussion

The prevalence of epilepsy in children with CP in this population-based cohort was 43% which is slightly higher than in an earlier Swedish study [18]. As previously described, the occurrence of epilepsy varied between different CP types [1], and it was more common in children with dyskinetic CP and less frequent in USCP [3].

Epilepsy in dyskinetic CP was associated with a more severe clinical picture, with infantile spasms and with CSWS. They had an early age of epilepsy onset, the highest frequency of status epilepticus, and many were on polytherapy. The occurrence of CSWS could be explained by the predominance of basal ganglia and thalamus lesions [19]. As reported previously, ID and more severe motor impairment were common [20]. Children with the most severe BSCP (tetraplegia) also had the highest seizure frequency and high use of polytherapy.

Neuroimaging findings depend on the timing of the brain insult [9]. The higher percentage of epilepsy seen in children born at term reflects their higher frequency of grey matter lesions, i.e., lesions acquired late in gestation. These results are in line with previous studies [21,22].

Both ID and more severe motor impairment were per se also associated with a higher frequency of epilepsy. A history of neonatal seizures, most common in dyskinetic CP, and the presence of severe ID and motor impairment represented a higher risk of developing epilepsy which is concurrent with other reports [23,24,25,26]. Symptomatic infantile spasms have been reported to be associated with a higher risk to develop other types of seizures and cognitive impairment [27]. Accordingly, almost all children with infantile spasms in our study developed other types of seizures and most had severe ID and motor impairment, related to the structural brain injury.

Epilepsy develops at an earlier age in children with CP compared to children without other neurological disorders [23]. In our study age at onset of epilepsy varied between the different CP types with earlier onset in children with the most severe BSCP, most likely due to more widespread brain pathology than in milder cases.

It is sometimes a challenge to differentiate epileptic seizures from other involuntary movements, especially in dyskinetic or ataxic CP [1]. Focal seizures were the most common seizure type in our cohort, but many children had multiple seizure types, especially those with dyskinetic CP and BSCP.

In our study the frequency of status epilepticus was 19% which was lower than in a previous Swedish study (47%) with children born 1987–1994 [18] which may be a result of improved treatment.

The most common finding on EEG was focal epileptic discharges also in children with generalized seizures. It is possible that some of these had a focal start although it had not been observed clinically. It is reasonable specifically to ask for a focal start in seizure semiology, as children with CP almost always have a structural cause of their epilepsy. Several children had multifocal epileptic discharges, most likely associated with the type and extent of underlying brain abnormality. Even though EEG is an important complement in the diagnosis of epilepsy it is important to remember that in children with CP, similar changes may be found even in the absence of clinical seizures [1].

The proportion of seizure free children had increased with age from 25% to 45% at the follow-up (at the age of 13–16 years), across all CP types. Similar findings have been reported by Tsubouchi et al. [7]. In the study by Tokatly Latzer et al. there was no difference in type of CP between the drug responsive or drug resistant epilepsy groups [6], while in another study remission rates varied by CP type, with the lowest remission rate in BSCP [28]. In our study, a better seizure outcome (seizure free for at least a year with or without ASM) was seen in girls, children with later epilepsy onset and predominant white matter injury, while children with epilepsy onset before 2 years of age, severe ID or severe motor impairment had a higher risk of persisting seizures. This is important to bear in mind when treating children with CP and epilepsy and when counselling their parents. There seemed to be a reluctance to discontinue ASM in seizure free children, probably due to the high risk of relapse, as reported by El Tantawi et al. [28]. However, we found that many children were on polytherapy despite seizure freedom for more than one year, often several years. We do not have the information if this was because attempts had been made to reduce ASM leading to relapses or if ASM was kept “to be on the safe side”. It may well be that at least the number of ASM could be reduced to the benefit of the children’s development, considering the possible side effects.

Valproate was the most common drug used in mono- and polytherapy and even if there was a slight decrease in its use in girls in 2019, there were still six girls in adolescence using valproate despite its known teratogenicity and risk for hormone abnormalities and polycystic ovarian syndrome [29].

Neuropsychiatric disorders such as ASD and ADHD are common in children with CP and are often associated with ID [8]. In this study we found an association between epilepsy and ASD but not with ADHD [30,31,32]. At the last follow-up we observed an increase in the diagnoses of both ID and ASD in children with CP and epilepsy but not in the group without epilepsy. This may be related to a negative impact of epilepsy and polytherapy on neurodevelopment as well as to the underlying brain impairment. In this cohort we have previously shown that the occurrence of intellectual disability, autism and ADHD increases with age as shown in repeated assessments [30]. Repeated assessments of cognitive function during childhood are important, in order to reveal these disabilities [33]. The background of the deterioration is probably multifactorial.

A strength of this study is the use of a population-based CP register including all children with CP born from 2003 to 2006 currently living in the county of Västra Götaland, Sweden, which eliminates selection bias. We reviewed all medical records from the paediatric clinics and habilitation centres comprising all possible places for the follow-up of children with CP.

The limitations of this study were its retrospective nature, and the fact that seizure semiology and frequency were not always well documented in the medical records.

## 5. Conclusions

In our population-based cohort of children with CP, 43% had epilepsy, all of them with onset before or at the age of 10 years. BSCP was the most common type of CP in children with epilepsy, while those with dyskinetic CP had the highest occurrence of epilepsy. Focal epilepsy dominated in all CP types. Epilepsy was associated with severe motor impairment, ID, and autism. Even if the epilepsy was often difficult to treat and many children were on polytherapy, 45% had been seizure free for more than one year by the age of 13–16 years. Prospective observational studies are warranted to identify cases who benefit from reduction or discontinuation of ASM.

## Figures and Tables

**Figure 1 jcm-12-00346-f001:**
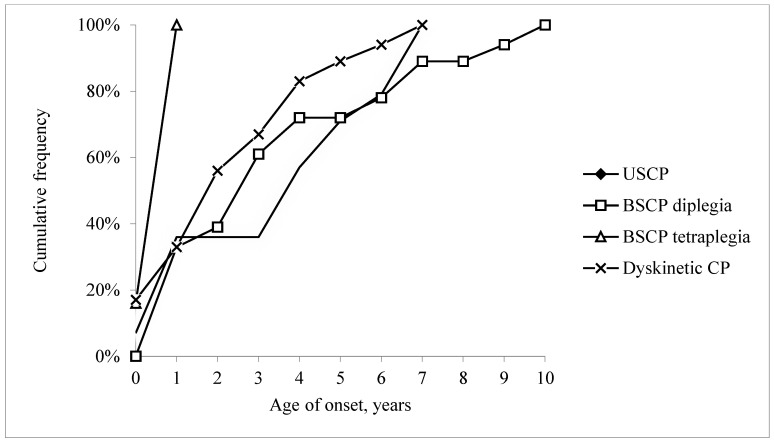
Cumulative frequency of epilepsy by age of onset (in years) in 59 children with epilepsy and unilateral spastic cerebral palsy (USCP), bilateral spastic cerebral palsy (BSCP; divided into diplegia and tetraplegia) and dyskinetic cerebral palsy.

**Table 1 jcm-12-00346-t001:** Characteristics of children with cerebral palsy (CP) with and without epilepsy.

	CP with Epilepsy	CP without Epilepsy	Total
	*n* = 59 (%)	*n* = 79 (%)	*n* = 138
Sex			
female	28 (47)	32 (53)	60
male	31 (40)	47 (60)	78
Gestational age			
≤28 weeks	5 (33)	10 (67)	15
28–31 weeks	5 (29)	12 (71)	17
32–36 weeks	6 (27)	16 (73)	22
≥37 weeks	43 (51)	41 (49)	84
Apgar score < 5 at 5 min			
≤36 weeks	3 (38)	5 (62)	8
≥37 weeks	12 (80)	3 (20)	15
Neonatal seizures ≤ 72 h			
Yes	27 (63)	16 (37)	43
No	32 (34)	63 (66)	95
Neuroimaging			
A. Maldevelopments	7 (64)	4 (36)	11
B. Predominant white matter injury	15 (28)	39 (72)	54
C. Predominant grey matter injury	25 (58)	18 (42)	43
C1. Basal ganglia/thalamus lesions	13 (65)	7 (35)	20
C2. Cortical-subcortical lesions only	8 (67)	4 (33)	12
C3. Arterial infarctions	4 (36)	7 (64)	11
D. Miscellaneous	11 (100)	0	11
E. Normal	1 (8)	12 (92)	13
Normal CT	0	2 (100)	2
Not done	0	4 (100)	4
Type of cerebral palsy			
USCP	14 (26)	39 (74)	53
BSCP	24 (46)	28 (54)	52
Dyskinetic CP	17 (71)	7 (29)	24
Ataxic CP	4 (44)	5 (56)	9
GMFCS level			
I	15 (24)	47 (76)	62
II	6 (29)	15 (71)	21
III	9 (60)	6 (40)	15
IV	12 (57)	9 (43)	21
V	17 (89)	2 (11)	19

Data are *n* (%). USCP; unilateral spastic cerebral palsy. BSCP; bilateral spastic cerebral palsy. GMFCS; Gross Motor Function Classification System.

**Table 2 jcm-12-00346-t002:** Characteristics of epilepsy in 59 children with cerebral palsy (CP).

	USCP	BSCP	Dyskinetic CP	Ataxic CP	Total
	*n* = 14	*n* = 24	*n* = 17	*n* = 4	*n* = 59 (%)
Type of seizures					
Focal	11	13	8	2	34 (57)
Generalized	1	3	1	2	7 (12)
Multiple types	2	8	7	0	17 (29)
Unknown	0	0	1	0	1 (2)
EEG: epileptic discharges (most recent EEG)					
Focal	10	15	8	1	34 (57)
Multifocal	1	7	7	1	16 (27)
Generalized	0	0	0	1	1 (2)
CSWS	1	0	0	0	1 (2)
Normal	2	2	2	1	7 (12)
Seizure frequency (last year)					
Seizure free	4	6	4	1	15 (25)
Seldom	7	4	5	1	17 (29)
Monthly	0	5	0	1	6 (10)
Weekly	0	1	1	0	2 (3)
Daily	1	5	2	0	8 (14)
Cluster	2	2	3	1	8 (14)
Unclear	0	1	2	0	3 (5)
Apgar score < 5 at 5 min	0	4	10	1	15 (25)
Neonatal seizures ≤ 72 h					
Yes	3	9	12	3	27 (46)
No	11	15	5	1	32 (54)
Infantile spasms					
Yes	0	2	6	0	8 (14)
No	14	22	11	4	51 (86)
Status epilepticus					
Yes	3	2	6	0	11 (19)
No	11	22	11	4	48 (81)
Antiseizure medication					
Monotherapy	5	13	7	2	27 (46)
Polytherapy	7	10	9	1	27 (46)
No ASM	2	1	1	1	5 (8)

Information is based on data obtained from medical records in 2015. *n*; number. CSWS; continuous spike-and-wave during sleep. ASM; antiseizure medication.

**Table 3 jcm-12-00346-t003:** Seizure frequency in 59 children with cerebral palsy and epilepsy at the age of 9–12 years (2015).

Type of Seizures	Seizure Free	Unclear	Seldom	Clusters	Monthly	Weekly	Daily	Total
Focal seizures	12	2	10	3	3	0	4	34
Generalised seizures	1	0	3	1	1	1	0	7
Multiple seizure types	1	1	4	4	2	1	4	17
Unknown	1	0	0	0	0	0	0	1
Total	15	3	17	8	6	2	8	59

## Data Availability

The data presented in this study are available on reasonable request from the corresponding author.

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
