# Peer review of "Characteristics and Challenges of Epilepsy in Children with Cerebral Palsy—A Population-Based Study"

_jcm, 2023, doi:10.3390/jcm12010346_

Round 1
Reviewer 1 Report
“Characteristics and challenges of epilepsy in children with cerebral palsy – a population-based study” is a descriptive article about the prevalence and characteristics of epilepsy in a cohort of children with cerebral palsy in 2015, based on the CP registry of Western Sweden, with a follow-up in 2019. The article length is justified by the amount of interesting information the authors have. You can find attached my suggestions.
· In the “Material and Methods” section, the authors state that “The medical records of all 140 children (61 girls and 79 boys) with CP born from 2003 to 2006…were reviewed”. A few sentences later, they write “Data were collected between July and October 2015” and then “The age range of the children at the time of the study was 9 to 12 years. A follow-up was done after 4 years, in 2019, at age 13 to 16 years”. Please could you confirm if this is the meaning of the abovementioned sentences: “The medical records of children with CP born from 2003 and 2006 were reviewed and demographic data (age, sex, etc.…) of the selected cohort were analyzed at two different timepoints: at ages 9-12 (data collected between July and October 2015) and 13-16 years (follow-up data collected in 2019)”? In case of affirmative answer, please simplify the text.
· I think that an important missing information could be the APGAR score at 5’. The authors cited a work in which a predictive model of drug-resistant epilepsy in CP patients is reported. The authors reported all the predictors (neonatal seizures, focal seizures and focal slowing on EEG), except for low APGAR at 5 minutes.
· Considering that the majority of patients had focal onset seizure, do you think there is a particular reason behind the fact that valproate, a broad-spectrum medication, is the most used ASM? Could it be related to a specific prescribing trend of ASMs in Sweden or there may be another reason?
· In paragraph 3.3 “Epilepsy treatment” a pie chart could be an easy way to summarize the types of ASMs taken by the patients.
· Line 227: There were particular reasons to stop or change valproate in the girls? Please specify the reasons (teratogenicity, adverse events, unknown,…).
· Lines 246-247: The authors stated: “Epilepsy in dyskinetic CP was associated with a more severe clinical picture, with infantile spasms and with CSWS. They had an early age of epilepsy onset, the highest frequency of status epilepticus, and many were on polytherapy”. If we carefully look at Table 2, zero patients with dyskinetic CP have CSWS. Is this a typo? If we consider the global clinical picture, BSCP seems to have a quite similar severe condition, considering the highest frequency of focal seizures and focal discharges in EEG, the highest seizure frequency (daily and cluster above all), the quite high number of neonatal seizures and the polytherapy.
· Lines 274-276: The authors stated: “The most common finding on EEG was focal epileptic discharges even in children with generalized seizures, most likely because the focal start had not been observed clinically. It is reasonable specifically to ask for a focal start in seizure semiology, as children with CP almost always have a structural cause of their epilepsy”. It is correct to think that structural lesions in the brain are associated with focal onset seizures rather than generalized ones. Maybe, people with CP have focal seizures, but witnesses describe them as generalized because they are not present from the start. However, having generalized seizures does not imply to always have generalized discharges in EEG. Atypical interictal EEG abnormalities can also be frequently encountered, including focal epileptiform discharges.
· Lines 293-295: The authors stated: “However, we found that many children were on polytherapy despite seizure freedom for more than one year, often several years, and it may well be that at least the number of ASM could be reduced to the benefit of the children’s development, considering the possible side effects”. I think it is impossible to know whether the seizure freedom is related to ASMs or to the natural course of epilepsy. Surely, EEG abnormalities and MRI/CT alterations increase the risk of seizure relapse, which seems high in this cohort of patients. However, reducing the pharmacological burden seems a reasonable choice, mostly in presence of adverse events. Was there someone between the polytherapy patients who reported adverse events? Do the authors know if an attempt to reduce ASM regimen was made and maybe a relapse was registered because of this?
· Lines 305-307 : “This may be related to a negative impact of epilepsy and polytherapy on neurodevelopment as well as to the underlying brain impairment. But it may also be due to repeated assessments of cognitive function during childhood, revealing these disabilities”. Please explain these sentences in a more exhaustive and detailed way, supported by scientific evidence.
· Line 314: “…the fact that seizure semiology and frequency were not always well documented in the medical records”. I would add in the Results section that it was not always possible to determine the frequency or semeiology of seizures due to not well documented medical records.
· Lines 316-318: “…we cannot tell whether the epilepsy was truly ‘drug resistant’ in cases with persisting seizures despite ASM. Moreover, those who were seizure free but on medication still by definition had epilepsy”. Please explain these sentences. What does “truly drug-resistant” mean? Do you imply there is a quote of pseudo-drug-resistant patients? If yes, why? Moreover, seizure free patients have epilepsy even though they achieve seizure control. Why should it be a limitation of the study?
· Lines 326-327: “Despite the risk of relapse due to the underlying brain pathology, a reduction or discontinuation of ASM appears to be an option in selected cases”. In which cases do the authors suggest the reduction/interruption of treatment?
· Lines 327-329 : “It is important to optimize ASM, to try to avoid polytherapy and to consider other treatment modalities like epilepsy surgery. There is a need for repeated assessments of cognitive function and neuropsychiatric comorbidities to optimize interventions”. Is this a perspective article? The conclusions seemed to be biased by the subjective point of view of the authors. The authors should be impartial, unless we are talking about a perspective article.
Reviewer 2 Report
This article discusses the epidemiology and clinical picture of epilepsy in childhood cerebral palsy, using a population-based sample from a region in Western Sweden. There have been a number of papers on the relationship between epilepsy and cerebral palsy. In this sense, there is little novelty in this paper. However, it uses a population database, which is a strength in terms of the reliability of the data. To make the paper clearer, the reviewers provide comments to the authors. Please read the comments and consider revising the paper.
(Minor1) How about adding to the introduction that there are few racial differences in the frequency of epilepsy?
(Miinor2) Please describe the definition of cerebral palsy clearly and properly based on literature11.
(Minor3) What is the paper's view on the presence or absence of intellectual disability in cerebral palsy?
(Minor4) Explain and describe the definition of CSWS in EEG.
(Minor5) Have there been any cases of epilepsy with a diagnosis of West syndrome?
(Minor6) Is it possible to compile a new table on the relationship between cerebral palsy and epilepsy that has been reported in the past?
(Minor7) Does the author have a theory as to why epilepsy is often combined with cerebral palsy?
(Minor8) Do you also have a theory as to why ADHD is more common?
Best regards
Dr. Reviewer
Round 2
Reviewer 1 Report
Thanks for accepting all the suggestions I gave. Good job!
Author Response
Dear Reviewer,
Thank you for accepting our revised manuscript! Minor revision of the english language has been done as suggested by reviewer 2.
Best regards
Ana Rufino
Reviewer 2 Report
Thank you for sending revised manuscript.
The purpose of this survey was to determine the prevalence and clinical characteristics of epilepsy in children with some types of cerebral palsy. With a focus on antiepileptic drugs and seizure outcome, using the registry of 140 CP cases born between 2003 and 2006 in western Sweden as basic population data.
In this context, the results, medical records of children aged 9-12 and 13-16 years were reviewed and 43% had a diagnosis of symptomatic epilepsy.
The authors answered almost all of the reviewers' questions and made revisions to the newly submitted manuscript. We appreciate that the west Swedish epidemiological clinical study, based on demographic statistics, is important from EBM as well as for planning future treatment other option including surgical strategies and replacement drugs. Finally, the authors should double-check the English text for any problems.
Best regards,
Dr. Reviewer
Author Response
Dear Reviewer,
Thank you for accepting our revised manuscript! Minor revision of the english language has been done as suggested by the reviewer.
Best regards
Ana Rufino